# DIFFUSION-BASED PHOTOREALISTIC BOKEH RENDERING FOR MOBILE DEVICES

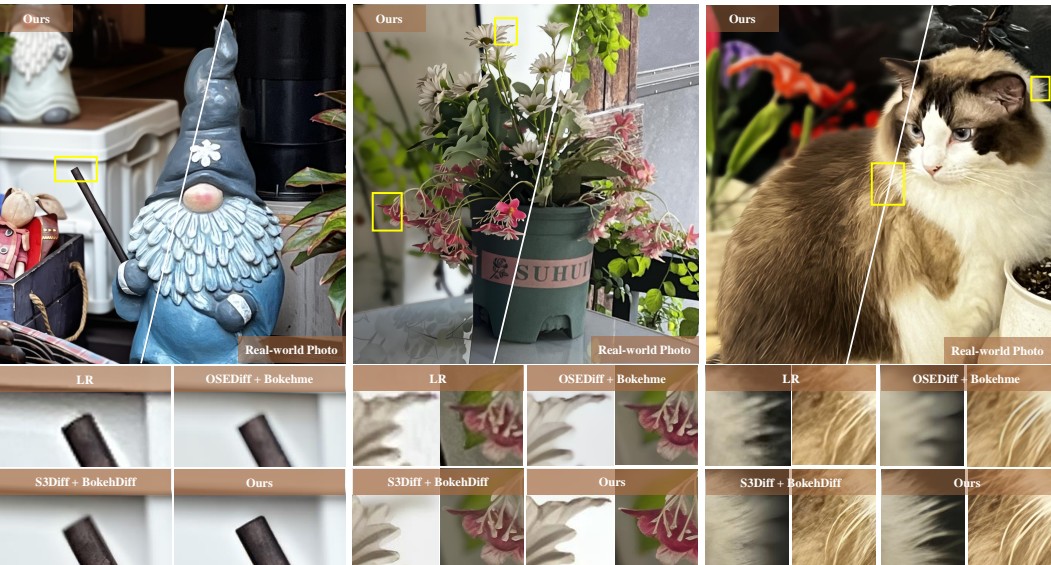

Figure 1: MagicBokeh is the first unified method specifically designed for high-zoom bokeh rendering in mobile devices. The inputs of real-world photos are images captured by an iPhone 13 at 5x digital zoom. (*Zoom-in for best view*)

## ABSTRACT

Photographs captured by mobile devices are often constrained by physical limitations, *i.e.*, small apertures, making it challenging to achieve the bokeh effects of shallow depth-of-field. Although previous work has primarily focused on learning-based methods to simulate bokeh effects for mobile images, they still face challenges when processing photos captured at high digital zoom levels on mobile devices, which often suffer from reduced resolution and degraded details. Therefore, it is still necessary to improve the quality of these inputs before creating the photorealistic bokeh effects, but this requirement will introduce inefficiencies in the workflow and lead to unnecessary error accumulation. To address the aforementioned issues, we propose MagicBokeh, a unified diffusion-based framework that improves both the quality and efficiency of bokeh rendering for high-zoom mobile photography. With the help of the proposed alternative training strategy and focus-aware mask attention, our approach achieves a unified optimization of bokeh rendering and super-resolution, thus improving both the controllability and quality of mobile bokeh rendering. Additionally, we further optimize depth estimation on low-quality images by degradation-aware depth module. Experiments demonstrate that MagicBokeh efficiently simulates high-quality bokeh effects under complex backgrounds, especially for digital zoom inputs from mobile devices. Code will be made publicly available.

# 1 INTRODUCTION

With the rapid advancement of mobile devices, smartphone photography has seen remarkable progress in recent years, which has greatly improved the photo-taking experience for users. However, limited by hardware constraints, current mobile devices often struggle to produce natural bokeh effects. Researchers have proposed many bokeh rendering methods which either rely on physical optics models to simulate light scattering (Kraus & Strengert, 2007; Lee et al., 2010; Wadhwa et al., 2018; Zhang et al., 2019; Sheng et al., 2024) or generate realistic bokeh effects by learning from large-scale datasets (Alzayer et al., 2023; Ignatov et al., 2020; Peng et al., 2022b; Wang et al., 2018). They can usually generate visually pleasing bokeh results and have been applied on mobile devices. Despite advances in these methods, one of the major limitations is that they all assume that the input is an all-in-focus high-quality (HQ) image. When applying these methods to images captured from a high digital zoom of the mobile camera, they often suffer from amplified noise, blurred subject boundaries, and unrealistic texture synthesis. Moreover, the quality degradation caused by digital zoom in mobile photography further hinders the effectiveness of existing bokeh rendering approaches, where the focused degraded regions usually affect the aesthetics.

To address this issue, a straightforward approach is using a two-stage pipeline: performing real-world image super-resolution (Real-ISR) first and then conducting bokeh rendering. However, such a naive approach results in two main problems: Firstly, since the output of the Real-ISR network is not always perfect, it may introduce error accumulation. These errors can be further amplified during the subsequent bokeh rendering process, ultimately degrading the overall image quality, as shown in Fig. 1. Secondly, the two-stage method requires two separate model inferences, which affects the computational efficiency on mobile devices. These limitations (shown in Fig. 2) naturally lead us to consider a unified approach.

Recently, diffusion models (Ho et al., 2020; Sohl-Dickstein et al., 2015; Song et al., 2020a;b), such as Stable Diffusion (SD) (Rombach et al., 2022), have demonstrated significant advantages in generating fine-grained image details and show remarkable generalization performance across various tasks, especially in Real-ISR. Moreover, we have observed that images produced by generative models often contain inherent bokeh information, indicating that these models possess the bokeh prior. This observation motivates us to consider whether we can design a unified diffusion-based approach that improves both the quality and efficiency of bokeh rendering for high digital zoom mobile devices.

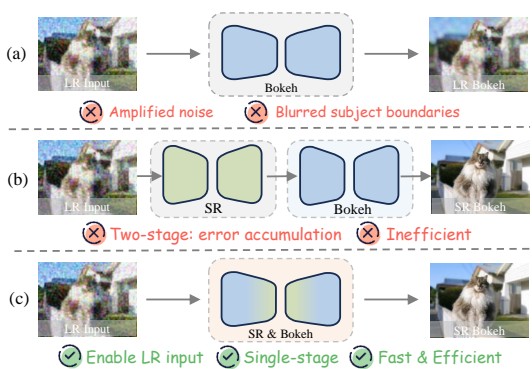

Figure 2: Compared with low-resolution (LR) bokeh rendering (a) and two-stage super-resolution (SR) bokeh rendering (b), our proposed method (c) seamlessly integrates the SR with bokeh rendering within a unified framework, thereby achieving both computational efficiency and photorealistic bokeh effects.

In this paper, we present MagicBokeh, a unified diffusion-based single-step framework designed for mobile photography that can efficiently generate bokeh effects for high-zoom photos. However, integrating Real-ISR and bokeh rendering into a unified model tends to introduce conflicting optimization objectives between two tasks, leading to performance degradation during training. To address this issue, we propose an alternative training strategy and focus-aware mask attention specifically designed for our framework.

To enhance computational efficiency, we compress the computationally intensive U-Net component in SD by well-designed block pruning. Depth Anything v2, with its powerful depth estimation capability, has been adopted as a depth prior in bokeh rendering tasks. Nevertheless, its performance is still challenged by image quality degradation. To address this issue, we propose a degradation-aware depth module, which improves the robustness and accuracy of depth estimation on low-quality (LQ) images. Experimental results demonstrate that our approach achieves valuable advancements in bokeh rendering for high-zoom mobile photography and also performs well in related tasks, such as refocusing. In summary, our main contributions are as follows:

- We propose MagicBokeh, a diffusion-based single-step framework that conducts Real-ISR and bokeh rendering simultaneously within a unified architecture.

- To further enhance high-zoom mobile bokeh rendering, we propose an alternative training strategy with focus-aware mask attention and introduce a degradation-aware depth module for improved depth estimation on high-zoom mobile photographs.

- Comprehensive experiments show that MagicBokeh achieves state-of-the-art (SOTA) quantitative and qualitative results on both synthetic and high-zoom real-world mobile photographs, highlighting its effectiveness in mobile photography.

## 2 RELATED WORKS

### 2.1 BOKEH RENDERING

Bokeh rendering refers to a computational photography technique that simulates the depth-of-field (DoF) effect. Existing bokeh rendering methods can be categorized into classical rendering methods and learning-based methods.

**Classical Rendering Methods.** Early bokeh rendering methods were primarily based on classical computer graphics, using ray tracing (Pharr et al., 2023; Potmesil & Chakravarty, 1981) to generate physically accurate bokeh effects. However, as the camera sampling space increased, the computational complexity increased exponentially, making these methods difficult to render fast. Subsequent methods improve efficiency by providing depth maps and focal plane information (Barron et al., 2015; Bertalmio et al., 2004; Soler et al., 2009; Wadhwa et al., 2018; Zhang et al., 2019; Busam et al., 2019). DeepFocus Senaras et al. (2018) specializes in using a perfect depth map to render realistic bokeh effects in low resolution. However, obtaining a perfect depth map in the real world is challenging. Dr.Bokeh Sheng et al. (2024) uses an inpainting model to estimate the RGBD values of occluded regions behind the salient object. It then simulates bokeh by computing the scattering and focusing of light in a spherical lens system based on foreground and background images, effectively reducing occlusion artifacts in boundary. Nevertheless, due to inaccuracies in the disparity maps, these methods often suffer from unnatural partial occlusion artifacts or color bleeding.

**Learning-based Methods.** Recent research has introduced neural rendering and generative models to address unnatural partial occlusion artifacts and color bleeding in bokeh rendering. BokehMe (Peng et al., 2022a) first generates bokeh effects using a classical physically motivated renderer and then employs a neural renderer to correct artifacts, mitigating the impact of imperfect disparity inputs. MPIB (Peng et al., 2022b) leverages an inpainting network to restore occluded background regions and applies an adaptive aggregation operation on a multiplane image layer, enabling the network to learn shallow DoF rendering across different focal planes. EBokehNet (Seizinger et al., 2023) integrates lens properties as additional inputs into the neural network to control the shape and intensity of the bokeh effect. BokehDiff (Zhu et al., 2025) is a diffusion-based lens blur rendering method that achieves physically accurate results with depth-aware attention. Despite recent advances, these methods still face significant challenges when applied directly to LQ inputs.

### 2.2 DIFFUSION-BASED REAL-ISR

Recent advances in generative diffusion models (Ho et al., 2020), particularly large-scale pre-trained text-to-image models (Rombach et al., 2022), have demonstrated exceptional performance in various downstream tasks, especially in ISR tasks (Lin et al., 2024; Xie et al., 2024; Moser et al., 2024; Yu et al., 2024). Recent studies have increasingly focused on single-step diffusion ISR models (Wang et al., 2024; Wu et al., 2024a; Zhang et al., 2024), which have shown great value when used on mobile devices. SinSR (Wang et al., 2024) presents a deterministic sampling technique that stabilizes the noise-image pair through consistency-preserving distillation. OSEDiff (Wu et al., 2024a) employs variational score distillation (Wang et al., 2023b) to maintain fidelity when generating high-resolution images. S3Diff (Zhang et al., 2024) leverages the T2I prior from SD-Turbo (Sauer et al., 2024) to achieve HQ images in a single step. Inspired by the aforementioned methods, we integrate the single-step Real-ISR task into the mobile bokeh rendering pipeline to enhance both generation quality and efficiency.

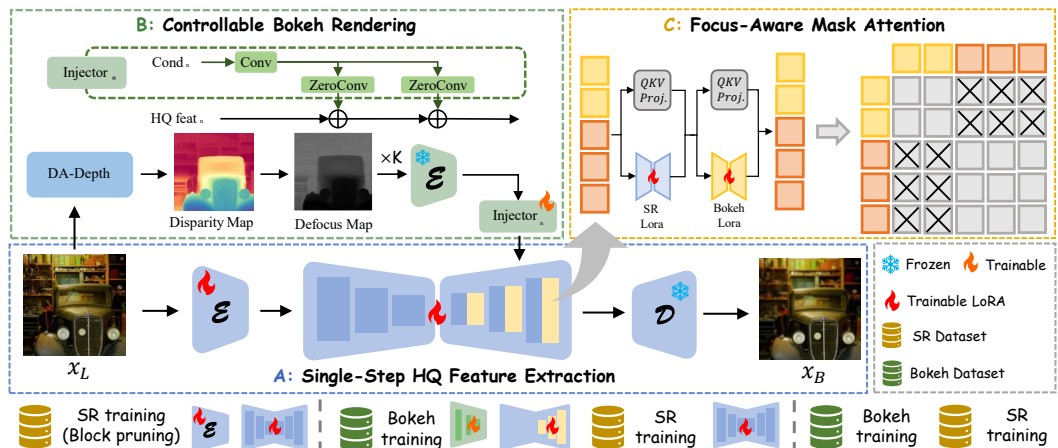

Figure 3: The framework of MagicBokeh. We introduce an alternative training strategy to unified Real-ISR and bokeh rendering together. During the bokeh training, the controlnet and bokeh LoRA layers are trainable to learn controllable bokeh rendering. During the Real-ISR training, only the SR LoRA is trainable to learn SR. During inference, given a high-zoom mobile photo, it can generate a disparity map through the degradation-aware depth model to guide bokeh rendering.

## 3 METHODOLOGY

### 3.1 FRAMEWORK OVERVIEW

Existing bokeh rendering methods based on generation models or lens blur rendering often rely on HQ image input, which is not suitable for high-zoom mobile photography. Therefore, we propose MagicBokeh, a diffusion-based framework that is highly suitable for this task while maintaining computational efficiency. As illustrated in Fig. 3, MagicBokeh consists of two main parts: HQ feature extraction and controllable bokeh rendering. The former extracts HQ features from LQ images, while the latter governs bokeh rendering based on the controllable bokeh rendering module and focus-aware mask attention.

**Single-Step HQ Feature Extraction.** Recent diffusion-based ISR approaches (Yue & Loy, 2024; Li et al., 2023a) have shown that directly using LQ images with little or no noise as input can substantially eliminate the uncertainty introduced by random noise sampling, while maximizing the retention of semantic content. Therefore, we directly feed the LQ images into the HQ feature extraction module without introducing any noise. Then, we inject Low-Rank Adaptation (LoRA) (Hu et al., 2022) into both the VAE encoder (only train in the first ISR training stage) and modified lightweight U-Net, and finetune the model to recover its HQ feature extraction capability. We employ L2 loss and LPIPS loss for supervision.

**Controllable Bokeh Rendering Module.** To achieve precise and controllable bokeh rendering, we introduce ControlNet (Zhang et al., 2023) as a conditional control module. In our framework, ControlNet receives a defocus map as the structural condition. Specifically, we first estimate a disparity map from the depth estimation network. The defocus map can be calculated by

$$r = K \left| d - d_f \right|, \tag{1}$$

where $d$ represents the disparity of the pixel, $d_f$ denotes the disparity of the focal position that the users specified, $K$ indicates the blur intensity, and $r$ represents the blur radius of the pixel. By integrating ControlNet, our model can generate visually plausible bokeh with controllable depth-of-field (DoF), while preserving semantic consistency in the in-focus regions.

### 3.2 ALTERNATIVE TRAINING STRATEGY

When implementing end-to-end training of our MagicBokeh framework using the SR bokeh dataset (containing paired LQ and HQ bokeh images), we observed a notable performance degradation in the ISR of subject areas, despite the original intention to simultaneously optimize both subject super

resolution and background bokeh rendering, as shown in the top part of Fig. 6. This degradation primarily arises from the conflicting optimization objectives inherent in Real-ISR and bokeh rendering tasks. Furthermore, the imbalance between the training samples for these tasks biases the network toward optimizing one task at the expense of the other.

To effectively address these challenges and mitigate conflict between tasks, we propose an alternative training strategy to decouple Real-ISR from bokeh rendering. This cyclical strategy alternates attention between different tasks. Initially, training emphasizes HQ bokeh rendering using LQ all-in-focus images as inputs, conditioned by defocus maps to generate HQ bokeh outputs. During this stage, the original diffusion model and pre-trained HQ feature extraction model are fixed, and training specifically targets the ControlNet and the bokeh LoRA layers in the focus-aware mask attention modules to refine the quality of bokeh rendering. Subsequently, training shifts to Real-ISR, employing pairs of LQ and HQ images as training samples. In this phase, optimization is restricted solely to the SR LoRA layers within the UNet of the diffusion network. We alternatively train these two phases. Our experiments validate that by alternating the focus between bokeh rendering and Real-ISR tasks, our proposed training strategy effectively reduces intertask interference, ultimately achieving significant improvements in the quality of mobile bokeh rendering.

### 3.3 Focus-aware Mask Attention

In our task, incorporating bokeh conditions directly into the generation process frequently results in degradation of the restoration quality for focused regions. To address this issue, we propose an approach that explicitly decouples Real-ISR from bokeh rendering, ensuring that the in-focus areas are accurately reconstructed without being affected by the defocused regions. Notably, in text-to-image models such as SD, self-attention layers play a crucial role in maintaining global coherence within generated images. Previous research (Epstein et al., 2023; Kim et al., 2023b) has shown that appropriately modulating self-attention layers can significantly enhance the controllability of generative results.

Although employing the alternative training strategy can alleviate conflicts between these two tasks, incorrect control still persists, particularly in image details. Consequently, we propose focus-aware mask attention, as shown in Fig. 3c, which utilizes focus cues obtained through the defocus maps as guidance for modulating self-attention layers. Specifically, we modulate the attention maps as below,

$$\text{Attention} = \text{softmax}\left(\frac{\mathbf{Q}\mathbf{K}^\top + \mathcal{M}}{\sqrt{d}}\right)\mathbf{V}, \tag{2}$$

where $\mathbf{Q}$, $\mathbf{K}$, $\mathbf{V}$ are the query, key and value of the self-attention layer, respectively. The focus attention mask $\mathcal{M}$ at feature location $(x, y)$ is

$$\mathcal{M}_{(x,y)} = \begin{cases} 0 & \text{if } M_{(x,y)} = 1 \\ -\infty & \text{otherwise} \end{cases}, \tag{3}$$

where $M$ is the binary result obtained by extracting the subject information from the defocus map at the focus region and binarizing the relationships between different regions (with the same regions being 1 and different regions being 0). This binary mask is resized to match the resolution required by the attention layer. During the training process, we alternately trained the SR LoRA layer and Bokeh LoRA layer. Notice that in the Real-ISR phase, the attention mask $\mathcal{M}$ is set to 0 to restore the whole image.

Integrating this attention mechanism into the proposed alternative training strategy enables a clear delineation of tasks: the ISR component is effectively directed toward prioritizing the focused subject area, whereas the bokeh rendering component is steered toward enhancing the background bokeh effects. This complementary interplay, guided by the defocus map, further facilitates achieving effective refocusing capabilities. Our experimental results demonstrate that the focus-aware mask attention module substantially enhances the controllability of our unified model, thus improving the overall quality of generated images.

### 3.4 Degradation-aware Depth Estimation

Despite the remarkable performance in HQ data, the accuracy of the depth estimation model deteriorates rapidly when applied to LQ images. The input of imperfect disparity map degrades the results

of SR bokeh rendering. To address this issue, we propose a self-feature distillation framework to estimate HQ-like features. We utilize the pre-trained Depth Anything v2 as the baseline network for both the teacher and student models. During the training process, both HQ images and simulated degraded images are respectively input into the teacher and student networks to extract features from encoder. Through feature distillation and output supervision, the features are expected to remain consistent, thereby improving the performance of depth estimation. Additional result analyses are provided in the Appendix. A.2.

## 4 EXPERIMENT

### 4.1 EXPERIMENTAL SETUP

**Training Datasets.** Following the setup of recent works (Wu et al., 2024b;a), we train our HQ feature extraction model on the LSDIR (Li et al., 2023b) and a subset of 10k face images from FFHQ (Karras et al., 2019). Additionally, to obtain HQ bokeh images as ground truth in bokeh training stage, similar to MPIB (Peng et al., 2022b) and Dr.Bokeh (Sheng et al., 2024), we built a ray-tracing-based renderer that generates lens blur through a real thin lens. More details are provided in the Appendix. A.3. During the training process, we use the degradation pipeline proposed in Real-ESRGAN (Wang et al., 2021) to synthesize the required LQ-HQ pairs. The synthesized LQ images are upscaled to match the HR resolution of $512 \times 512$ before feeding into our model.

**Evaluation Metrics.** We evaluate the performance of various methods using both full-reference and no-reference metrics. First, we use PSNR, SSIM (Wang et al., 2004), and LPIPS (Zhang et al., 2018) to measure the fidelity of the bokeh rendering. We also use reference-based perceptual metrics such as DISTS (Ding et al., 2020), image generation similarity metrics like FID (Heusel et al., 2017), and no-reference metrics including NIQE (Zhang et al., 2015), MANIQA (Yang et al., 2022), MUSIQ (Ke et al., 2021), and CLIPIQA (Wang et al., 2023a).

**Implementation Details.** Our single-step HQ feature extraction model is built upon SD2.1, where we remove all cross-attention layers and the mid-stage module in the original U-Net by well-designed block pruning. Specifically, through experimental observations on existing single-step Real-ISR methods (Wu et al., 2024a; Zhang et al., 2024), we observe that while text prompts provide semantic information, they offer limited benefits and significant computational overhead in extracting HQ features in practice. Consequently, we remove the text encoder and cross-attention modules from the pipeline, effectively eliminating prompt dependency and reducing computational overhead. Following (Kim et al., 2023a), we streamline the U-Net architecture by removing the entire mid-stage module, which significantly improves efficiency without compromising perceptual quality. Then we inject LoRA modules into both the VAE encoder and the modified lightweight U-Net, and retrain the model on the Real-ISR dataset with paired LQ-HQ images. We use the AdamW optimizer with a learning rate set to $5e$-5. Then, we adopt an alternative training strategy consisting of two phases. In the bokeh rendering phase, we train the controllable bokeh rendering module and the bokeh LoRA layers in the focus-aware mask attention module on the SR bokeh dataset containing paired LQ and HQ bokeh images. We use the AdamW optimizer with a learning rate set to $5e$-5. In the following Real-ISR phase, we train the SR LoRA layers of the UNet on the ISR dataset, employing the AdamW optimizer with a learning rate of $5e$-6. The entire training process takes approximately 20 hours on 4 NVIDIA L40 GPUs. In addition, we apply random horizontal flipping to enhance the diversity of the training data.

### 4.2 RESULTS ON SYNTHETIC DEGRADATION DATASET

**Synthetic Degradation Dataset.** We conduct a systematic evaluation of bokeh rendering performance on the real-world EBB dataset. For the established EBB400 benchmark, we randomly select 400 image pairs and manually label the focal regions in each image to assess the bokeh rendering accuracy. We applied this benchmark to evaluate images in the high-zoom mobile bokeh rendering task, named EBB400-LQ, where image degradation was simulated using the Real-ESRGAN pipeline. To ensure a fair comparison, since the compared methods obtain SR images after the first stage, Depth Anything v2 is used to generate disparity maps. In contrast, our approach employs the proposed degradation-aware depth network to directly estimate more robust disparity maps from the original LQ inputs. All disparity maps are normalized to 0-1 during testing to ensure consistency.

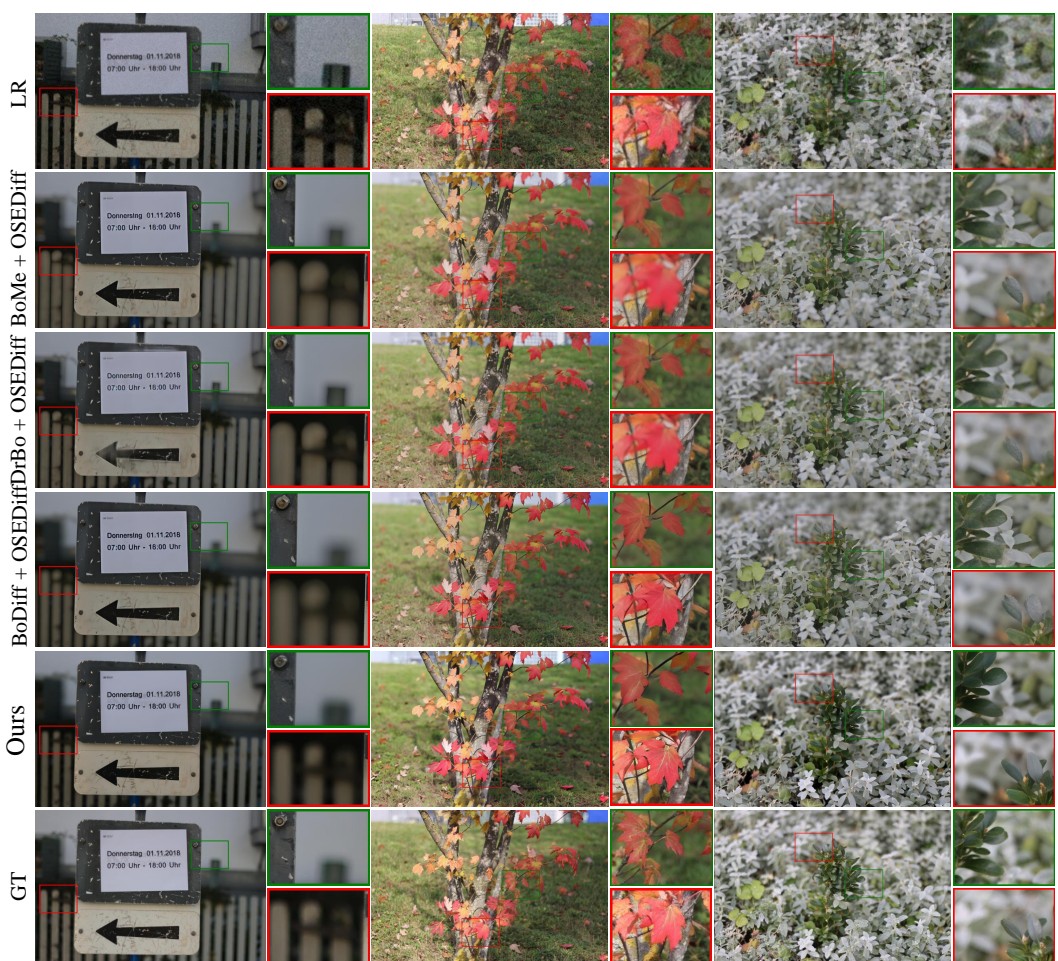

Figure 4: Qualitative comparison on EBB400-LQ. More can be seen in the Appendix. A.5.

Table 1: Quantitative comparison of performance with two-stage SOTA models on EBB400-LQ benchmark. ISR methods use OSEDiff (*) and S3Diff($^+$). The inference times are tested with an input image of size $512 \times 512$, and the inference time is measured on an L40s GPU. **Bold** and underline denote the best and the second best result.

| Method | PSNR ↑ | SSIM ↑ | LPIPS ↓ | DISTS ↓ | MUSIQ ↑ | MANIQA ↑ | FID ↓ | Time(s) ↓ |
|---|---|---|---|---|---|---|---|---|
| BokehMe* | 23.51 | 0.8459 | 0.3106 | 0.1666 | 57.70 | 0.4219 | 72.98 | 0.1648 |
| Dr.Bokeh* | 23.39 | 0.8488 | 0.3132 | 0.1677 | 52.40 | 0.3934 | 73.38 | 2.4021 |
| BokehDiff* | 23.65 | 0.8459 | 0.3049 | 0.1713 | 59.24 | 0.4251 | 72.65 | 0.3376 |
| BokehMe$^+$ | 23.75 | 0.8388 | 0.3138 | 0.1606 | 57.54 | 0.4137 | **72.25** | 0.7510 |
| Dr.Bokeh$^+$ | 23.67 | 0.8430 | 0.3134 | 0.1687 | 52.63 | 0.3876 | 73.10 | 2.9883 |
| BokehDiff$^+$ | 23.83 | 0.8397 | 0.3071 | 0.1735 | **59.36** | **0.4259** | 72.54 | 0.9238 |
| MagicBokeh | **24.23** | **0.8623** | **0.2786** | **0.1600** | 58.83 | 0.4138 | 72.43 | **0.1062** |

**Quantitative Experiment.** To validate the effectiveness of our method, we compare MagicBokeh with two-stage pipelines, including SOTA diffusion-based Real-ISR methods and bokeh rendering methods. Specifically, considering that recent work has focused mainly on the diffusion-based single-step framework, we evaluate our method against Real-ISR methods including OSEDiff (Wu et al., 2024a), and S3Diff (Zhang et al., 2024). The approaches which require depth maps always get better bokeh effects, so we compare with bokeh rendering methods including BokehMe (Peng et al., 2022a), Dr.Bokeh (Sheng et al., 2024) and BokehDiff (Zhu et al., 2025). As shown in the Tab. 1, our model achieves SOTA performance compared to previous two-stage SOTA methods, demonstrating its superior effectiveness in high digital zoom mobile bokeh rendering. Although our method performs worse than BokehDiff in some non-parameterized metrics, this is due to BokehDiff generating more focused areas in the EBB400-LQ dataset, leading to higher metric values. However, this im-

provement is not realistic in terms of the actual bokeh effect. Therefore, we performed a qualitative comparison in Fig. 4. The conclusion is further validated by the visual comparison in Appendix. A.5. Through this comparison, we observe clear limitations in the performance of other two-stage methods. Firstly, the two-stage methods require two separate model inferences, which lead to inefficiency. Moreover, the edge artifacts introduced during Real-ISR lead to bokeh rendering with unnatural edge transitions. In contrast, by reusing the prior information from the diffusion model and adopting an alternative training strategy along with focus-aware mask attention, our approach delivers superior bokeh quality and computational efficiency compared to other methods. Bokehme struggles to render natural partial occlusion. Dr.Bokeh incorporates an inpainting model, which introduces more artifacts and blurriness. OSEDiff and S3Diff rely on semantic conditions as inputs, which often leads to semantic errors in the Real-ISR process. Furthermore, although we do not use any text conditions, MagicBokeh still shows strong performance in the single task of Real-ISR compared with single-step Real-ISR methods, as illustrated in the Appendix. A.4, highlighting its ability to restore both the realism and aesthetic quality of images.

## 4.3 User study on Real-world Dataset

**Real-world Degradation Dataset.** Synthetic degradation datasets fail to capture the complex artifacts in real-world mobile photography, such as hybrid sensor circuit noise, motion blur from handheld shooting and lossy compression in digital zoom. To address this, we design a user study specifically on authentic LQ images captured under practical mobile photography conditions. We collected 50 real-world LQ images using an iPhone 13 pro, covering diverse scenarios (portraits, landscapes, indoor/outdoor scenes) with varying high digital zoom levels ($5\times - 15\times$). The average resolution of the images is $4032 \times 3024$.

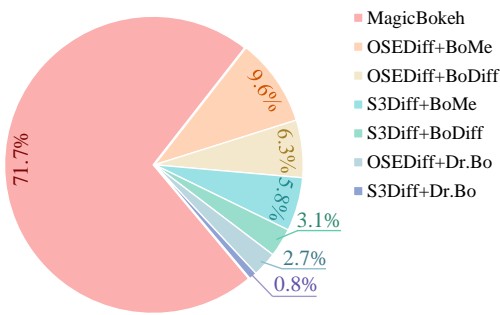

Figure 5: The figure shows the human preference.

**Quantitative Results.** This study engages 50 participants from diverse backgrounds, ensuring a wide range of perspectives. Each participant is presented with bokeh images from different methods, and they are then asked to choose the best one from these images. As shown in Fig. 5, our method achieves outstanding scores compare with other two-stage approaches in the HQ bokeh rendering task for high-zoom mobile photography.

## 4.4 Ablation Studies

In this section, we perform a comprehensive ablation study to assess the impact of each component in MagicBokeh on the EBB400-LQ dataset.

Table 2: Ablation study on the EBB400-LQ dataset. The setting of "FAMA", "Strategy", and "DA depth" are short for the focus-aware mask attention, alternate training strategy, and degradation-aware depth module respectively. **Bold** and underline denote the best and the second best result.

| Modules | | | Metrics | | | | | | |
|---|---|---|---|---|---|---|---|---|---|
| FAMA | Strategy | DA depth | PSNR ↑ | LPIPS ↓ | CLIP-IQA ↑ | NIQE ↓ | MUSIQ ↑ | MANIQA ↑ | FID ↓ |
| ✗ | ✗ | ✗ | 24.21 | 0.2931 | 0.3743 | 6.0786 | 57.41 | 0.4038 | 73.25 |
| ✗ | ✓ | ✓ | 24.22 | 0.2798 | 0.4157 | 5.9068 | 58.10 | 0.4065 | 75.23 |
| ✓ | ✗ | ✓ | 24.20 | 0.2946 | 0.3781 | 5.7076 | 56.08 | 0.3956 | 73.04 |
| ✓ | ✓ | ✗ | 24.20 | **0.2784** | 0.4209 | 5.8035 | 58.80 | 0.4114 | 75.03 |
| ✓ | ✓ | ✓ | **24.23** | 0.2786 | **0.4229** | **5.6341** | **58.83** | 0.4138 | **72.43** |

**Effect of Focus-aware Mask Attention.** To validate whether focus-aware mask attention can effectively reconstruct the focal region while being unaffected by the defocused areas, we designed a single-contrast variant model, referred to as *w/o* focus-aware mask attention (*w/o* FAMA). This variant model does not use the focal cues obtained from the defocused image to modulate self-attention,

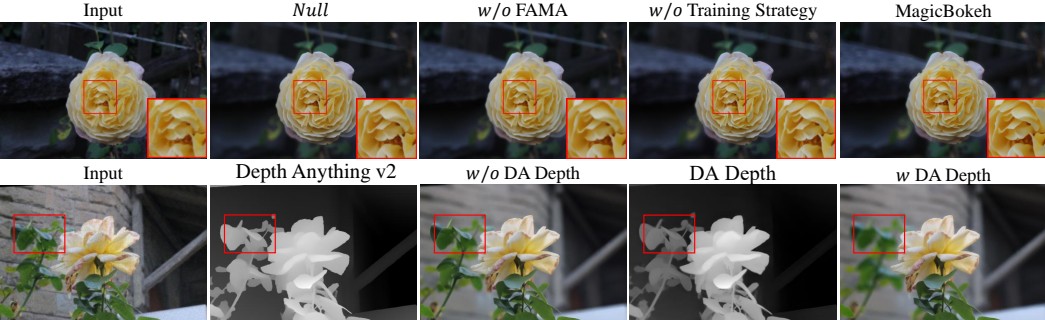

Figure 6: Visual comparison of the ablation study.

instead applying attention operations only on the global image. The results are shown in Tab. 2. As can be seen, the full model and *w/o* FAMA seems minor differences in PSNR, the no-reference metrics show significant improvement in the full model. This indicates that the focus-aware mask attention mechanism can successfully decouple the focused subject from the out-of-focus area.

**Effect of Alternate Training Strategy.** To verify whether the alternate training strategy improves the quality of subject Real-ISR and the blurring effect of the defocus region, we designed another single contrast variant model, named *w/o* alternative training strategy (*w/o* Strategy). The results are shown in Tab. 2 and Fig. 6. As can be seen, compared to *w/o* strategy, the full model, which includes the alternative training strategy, enhancing the quality of bokeh rendering.

**Effect of Degradation-Aware Depth Module.** To assess the contribution of degradation-aware depth module in MagicBokeh, we input the disparity map predicted by Depth Anything v2 into the network and conduct a comparative experiment, named *w/o* DA depth. To verify, the results are listed in Tab. 2. Although there is no substantial difference in quantitative metrics between *w/o* DA depth and our method, we can find improvement in qualitative comparison, as shown in bottom part of Fig. 6. DA depth provides better depth estimation results for LQ images.

## 4.5 FURTHER APPLICATION

While existing bokeh rendering methods assume all-in-focus inputs, mobile photograph often contain partially defocused regions due to autofocus errors or multi-subject compositions. Thus, reconstructing sharp image areas that are blurred by the bokeh effect and refocusing on new regions of interest presents a critical challenge. Our method, which is built upon LQ input images, is found to generalize well to the task of refocusing. As shown in Fig. 7, the result demonstrate that our approach significantly produces smooth blur transitions when shifting focus from the coffee cup to background chairs.

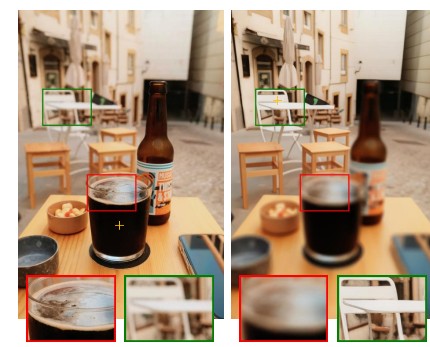

Input (Focus on coffee) Output (Refocus on chair)

Figure 7: Further application in refocusing.

## 5 CONCLUSION

In this paper, we present MagicBokeh, a unified diffusion-based framework for high-zoom bokeh rendering on mobile devices. Our method jointly performs Real-ISR and bokeh rendering in a unified architecture, thereby effectively overcoming the limitations of traditional two-stage pipelines. To address the conflicting objectives between image super-resolution and bokeh rendering, we introduce an alternating training strategy that enables the model to learn both tasks efficiently. Furthermore, we design two plug-and-play modules, namely controllable bokeh rendering and focus-aware mask attention, to guide bokeh rendering and enhance subject-background separation, respectively. And we also propose well-designed block pruning to further optimize computational efficiency. Extensive experiments and a user study demonstrate that MagicBokeh achieves SOTA results in high-zoom mobile bokeh rendering and is well suited for real-world mobile photography applications.

ETHICS STATEMENT

This study is based on publicly available datasets under their respective licenses. No new data involving human subjects were collected. All visualizations respect privacy. We confirm that our methods and experiments do not raise additional ethical concerns.

REPRODUCIBILITY STATEMENT

We use publicly accessible datasets, LSDIR (Li et al., 2023b), FFHQ (Karras et al., 2019), NYUv2 (Silberman et al., 2012), KITTI (Geiger et al., 2013), RealSR (Cai et al., 2019) , DrealSR (Wei et al., 2020), PhotoMatte85 (Lin et al., 2021), RWP-636 (Yu et al., 2021), AIM-500 (Li et al., 2021) and SA-1B (Kirillov et al., 2023). After the blind review period, we will release our codebase, training/inference scripts, configuration files, and model checkpoints, together with step-by-step instructions and evaluation protocols to fully reproduce all results.

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

# A  APPENDIX

In the appendix, we first provide the LLM usage statement, ethics statement, and reproducibility statement. We then present a detailed description of Degradation-aware Depth Estimation and the pipeline of the bokeh training dataset. Subsequently, we report additional experimental comparisons, including quantitative comparison on Real-ISR and qualitative comparison on real-world photos.

**Appendix is organized as follows:**

CONTENTS

## A.1  USE OF LLMS

The LLMs are used only for language polishing and editing of the manuscript text.

## A.2  DEGRADATION-AWARE DEPTH ESTIMATION

### A.2.1  TRAINING DETAILS

Despite the remarkable performance in HQ data, the accuracy of the depth estimation model deteriorates rapidly when applied to LQ images. And the input of imperfect disparity map degrades the results of SR bokeh rendering.

To address this issue, we propose a self-feature distillation framework to estimate HQ-like features. As shown in Fig. 8, we utilize the pre-trained Depth Anything v2 Large model as the baseline network for both the teacher and the student models. During the training process, both HQ images and simulated degraded images are respectively input into the teacher and student networks to extract features from encoder. Through feature distillation, features are expected to remain consistent, thereby improving depth estimation performance. Simultaneously, the network's output is supervised to obtain a more accurate depth map.

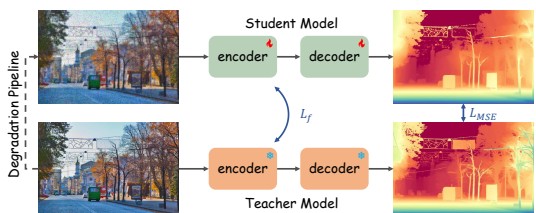

Figure 8: The training pipeline of the degradation-aware depth module.

### A.2.2  QUANTITATIVE COMPARISON OF DEPTH ESTIMATION

In our experiments, we use the pre-trained Depth Anything v2 as the teacher model to generate pseudo-labels and supervise the student model, initialized identically, within a distillation framework that takes only RGB images as input. Specifically, we conduct our distillation experiments using a subset of 200,000 samples from the SA-1B dataset (Kirillov et al., 2023). The Real-ESRGAN degradation pipeline (Wang et al., 2021) is used to synthesize LQ-HQ training pairs.

To demonstrate the effectiveness of our degradation-aware depth model on degraded images, we compare our approach with the baseline method, Depth Anything v2. Tab. 3 shows that our method outperforms these related works on the degraded NYUv2 (Silberman et al., 2012) (for indoor scenes)

Table 3: Quantitative comparison on the NYUv2 and KITTI datasets (seen datasets with synthetic degradations) for "Degrade", "Clear", and "Average" scenarios.

| Dataset | Method | Degrade | | Clear | | Average | |
|---|---|---|---|---|---|---|---|
| | | AbsRel ↓ | $\delta_1$↑ | AbsRel↓ | $\delta_1$↑ | AbsRel↓ | $\delta_1$↑ |
| NYUv2 | Depth Anything v2 | 0.081 | 0.926 | **0.043** | **0.981** | 0.062 | 0.954 |
| | DA depth | **0.068** | **0.946** | 0.047 | 0.976 | **0.058** | **0.961** |
| KITTI | Depth Anything v2 | 0.123 | 0.852 | **0.074** | **0.946** | 0.099 | 0.899 |
| | DA depth | **0.105** | **0.883** | 0.079 | 0.944 | **0.092** | **0.914** |

and KITTI (Geiger et al., 2013) (for outdoor scenes). We use point prompts for "Degrade", "Clear" and "Average" scenarios. "Degrade" refers to images degraded by Real-ESRGAN, "Clear" refers to the original, non-degraded images, and "Average" is the mean value of the "Degrade" and "Clear" images. The bold values indicate the best performance. Through self-feature distillation, our student model not only exhibits minimal performance degradation on clear images but also outperforms the baseline on degraded images, thereby verifying the superiority of our method.

## A.3 DETAIL OF BOKEH TRAINING DATASETS

To obtain HQ bokeh images as ground truth in bokeh training stage, similar to MPIB (Peng et al., 2022b) and Dr.Bokeh (Sheng et al., 2024), we built a ray-tracing-based renderer that generates lens blur through a real thin lens, as shown in Fig. 9. We first collected nearly 2k high-resolution landscape images from the Internet to serve as our background images. The foreground images are collected from PhotoMatte85 (Lin et al., 2021), RWP-636 (Yu et al., 2021), AIM-500 (Li et al., 2021) and websites. Each sample is randomly composed

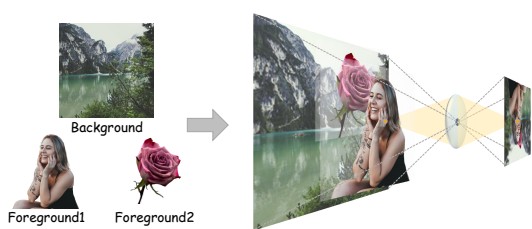

Figure 9: The pipeline of data synthesis.

of two selected foreground images and one background image. During the composition process, the disparity map is set within the range from 0 to 1, the random blur parameter ranges from 0 to 32, and the disparity focus is randomly set to one of the positions in either the foreground or the background. In order to introduce more variation in depth and create more diverse blur effects in the training data, we randomly set the depth variation for the background.

Table 4: Quantitative comparison with state-of-the-art methods on real-world benchmarks (RealSR (Cai et al., 2019) and DrealSR (Wei et al., 2020)). By providing a defocus map with all-zero input, our method can generate a high-quality all-in-focus image for quantitative comparison. The best and second-best results are highlighted in **red** and blue.

| Datasets | Methods | PSNR ↑ | SSIM ↑ | LPIPS ↓ | DISTS ↓ | CLIP-IQA ↑ | NIQE ↓ | MUSIQ ↑ | MANIQA ↑ | FID ↓ |
|---|---|---|---|---|---|---|---|---|---|---|
| RealSR | SinSR | **26.32** | 0.7363 | 0.3195 | 0.2351 | 0.6153 | 6.3541 | 60.42 | 0.5366 | 138.64 |
| | OSEDiff | 25.15 | 0.7341 | 0.2921 | 0.2128 | 0.6685 | 5.6528 | **69.11** | 0.6332 | 123.68 |
| | S3Diff | 25.18 | 0.7269 | **0.2722** | **0.2005** | **0.6742** | **5.2612** | 67.82 | **0.6417** | **105.11** |
| | MagicBokeh | 26.14 | **0.7392** | 0.2888 | 0.2192 | 0.6246 | 5.6337 | 67.22 | 0.6214 | 123.06 |
| DRealSR | SinSR | 28.35 | 0.7484 | 0.3689 | 0.2497 | 0.6319 | 6.9533 | 55.09 | 0.4881 | 170.18 |
| | OSEDiff | 27.91 | 0.7834 | **0.2968** | 0.2269 | 0.6964 | 6.4907 | **64.65** | 0.5899 | 135.28 |
| | S3Diff | 27.54 | 0.7491 | 0.3109 | **0.2100** | **0.7132** | 6.1935 | 63.93 | **0.6099** | **118.57** |
| | MagicBokeh | **28.99** | **0.7901** | 0.3003 | 0.2220 | 0.6633 | **6.1204** | 62.93 | 0.5901 | 143.02 |

## A.4 QUANTITATIVE COMPARISON ON REAL-ISR

Although our method is not specifically designed for super-resolution tasks, setting the blur intensity $K$ to 0 allows us to obtain all-in-focus HR images. Furthermore, despite not incorporating text conditions, MagicBokeh still shows performance in the single task of Real-ISR, as illustrated in Tab. 4, highlighting its ability to restore both the realism and aesthetic quality of images.

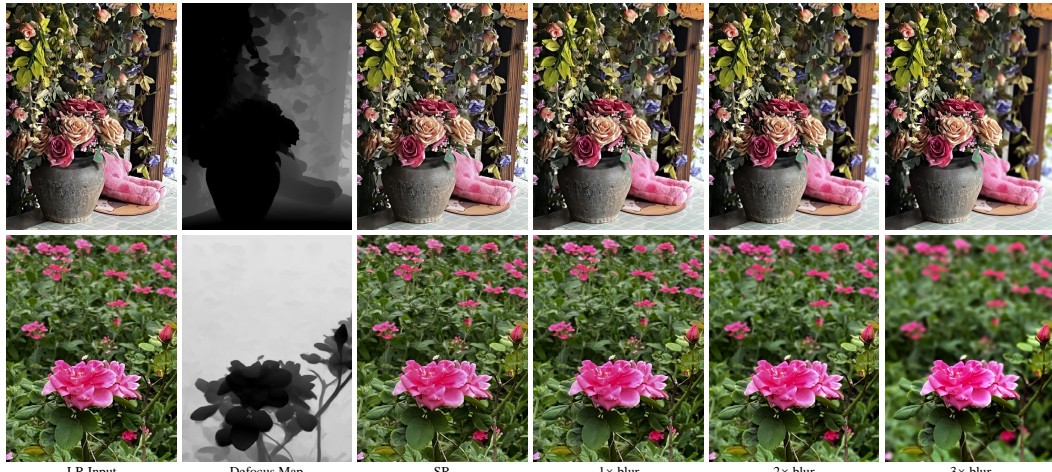

| | | | | | |
|---|---|---|---|---|---|
| LR Input | Defocus Map | SR | 1× blur | 2× blur | 3× blur |

Figure 10: Given the defocus map and LR input, our method is able to gradually increase the aperture parameter from 1x blur to 3x blur (*Zoom-in for best view*).

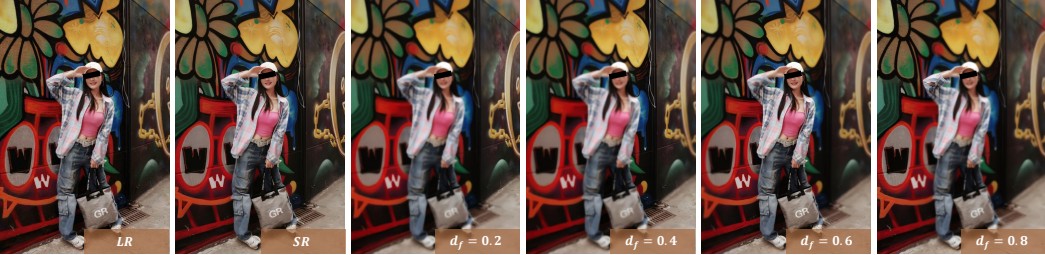

Figure 11: Given the disparity map and LR input, our method is able to achieve dynamic adjustment of the focus distance (*Zoom-in for best view*).

## A.5 MORE RESULTS

**Adjusting Aperture.** We present the results of increased blurriness in Fig. 10. MagicBokeh successfully achieves progressive blurriness while maintaining subject sharpness. The cases are high-zoom real mobile device captures, and MagicBokeh generates realistic bokeh effects.

**Adjusting Focus Distance.** We provide examples of changing focus distance in Fig. 11. Whether focusing on the foreground or background, our method can achieve natural super-resolution and bokeh effects.

**More Comparisons.** Here, we provide more comparisons between MagicBokeh and other two-stage pipeline to further validate the effectiveness of MagicBokeh.

First, we demonstrate more comparisons in Fig. 12. In the first example, MagicBokeh produces bokeh effects that are closer to the Ground Truth compared to other methods, especially in the red-boxed area. Compared to methods including BokehMe and Dr.Bokeh in the green-boxed area, our method and BokehDiff generate sharper edges. In the second example, in terms of super-resolution, our method produces more distinct leaf details compared to OSEDiff and S3Diff. In terms of bokeh, our method generates the best edge effects compared to BokehDiff, BokehMe, and Dr.Bokeh. In the third example, our method can still produce bokeh effects that are consistent with the real situation, even in the presence of noise. We continue the demonstration of results in Fig. 13. Our method gradually increases the blur with increasing defocus while keeping the focused foreground unchanged, resulting in a more realistic effect. We present more results in Fig. 14, derived from 5x zoom images captured with an iPhone 13, demonstrating that our method is well suited for real-world mobile photography applications.

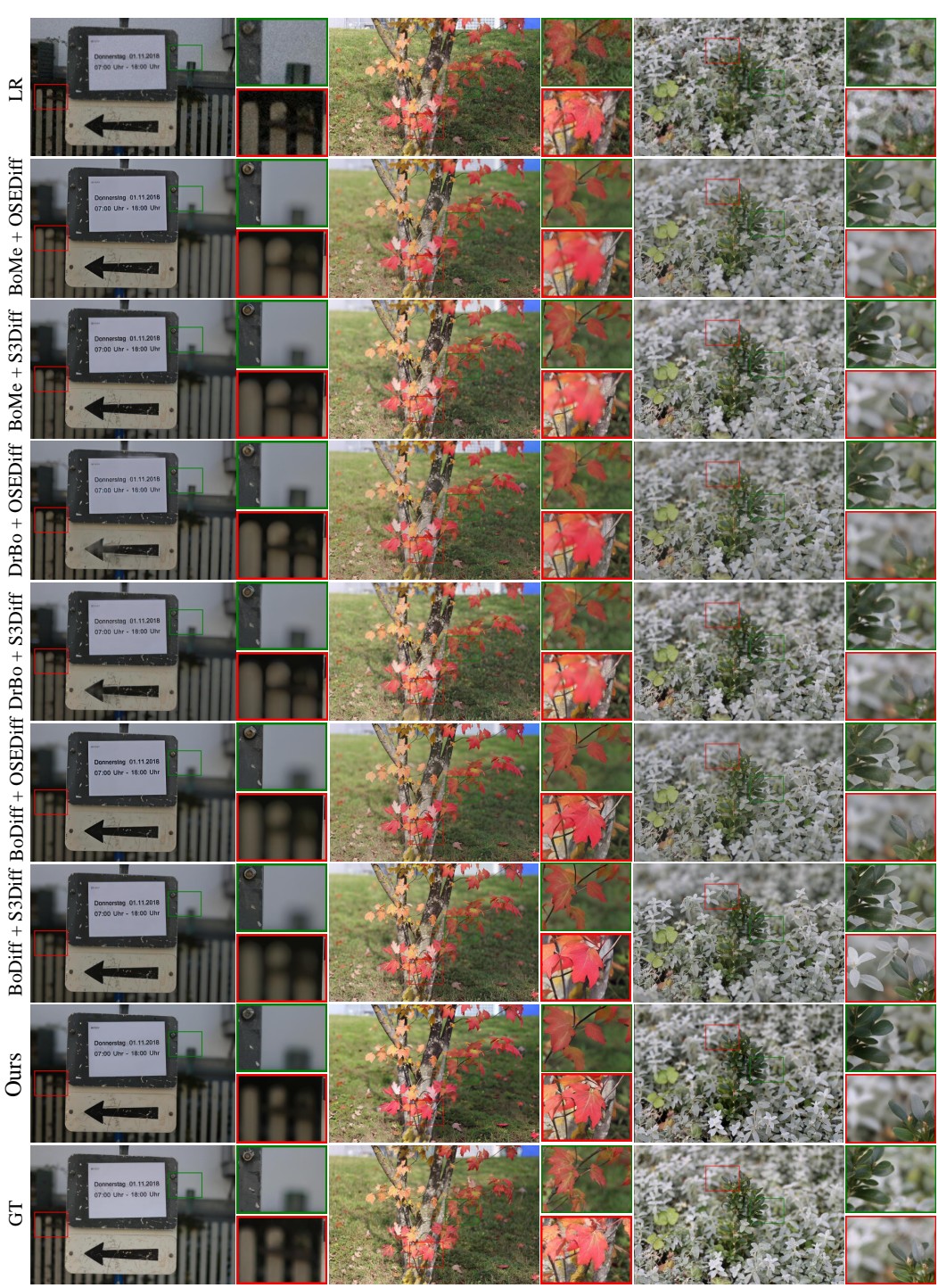

Figure 12: (Qualitative comparison on EBB400-LQ *Zoom-in for best view*).

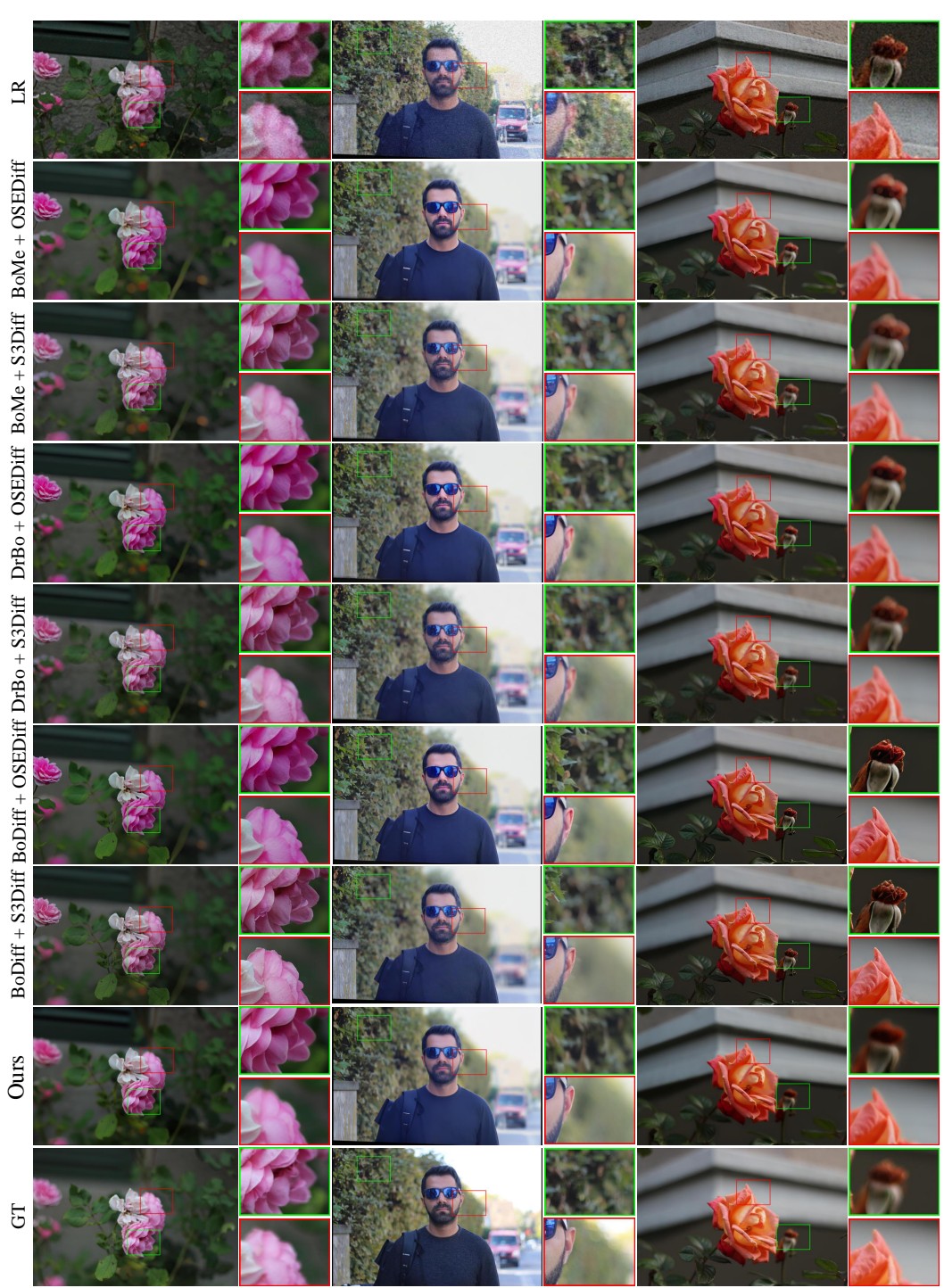

Figure 13: Qualitative comparison on EBB400-LQ (*Zoom-in for best view*).

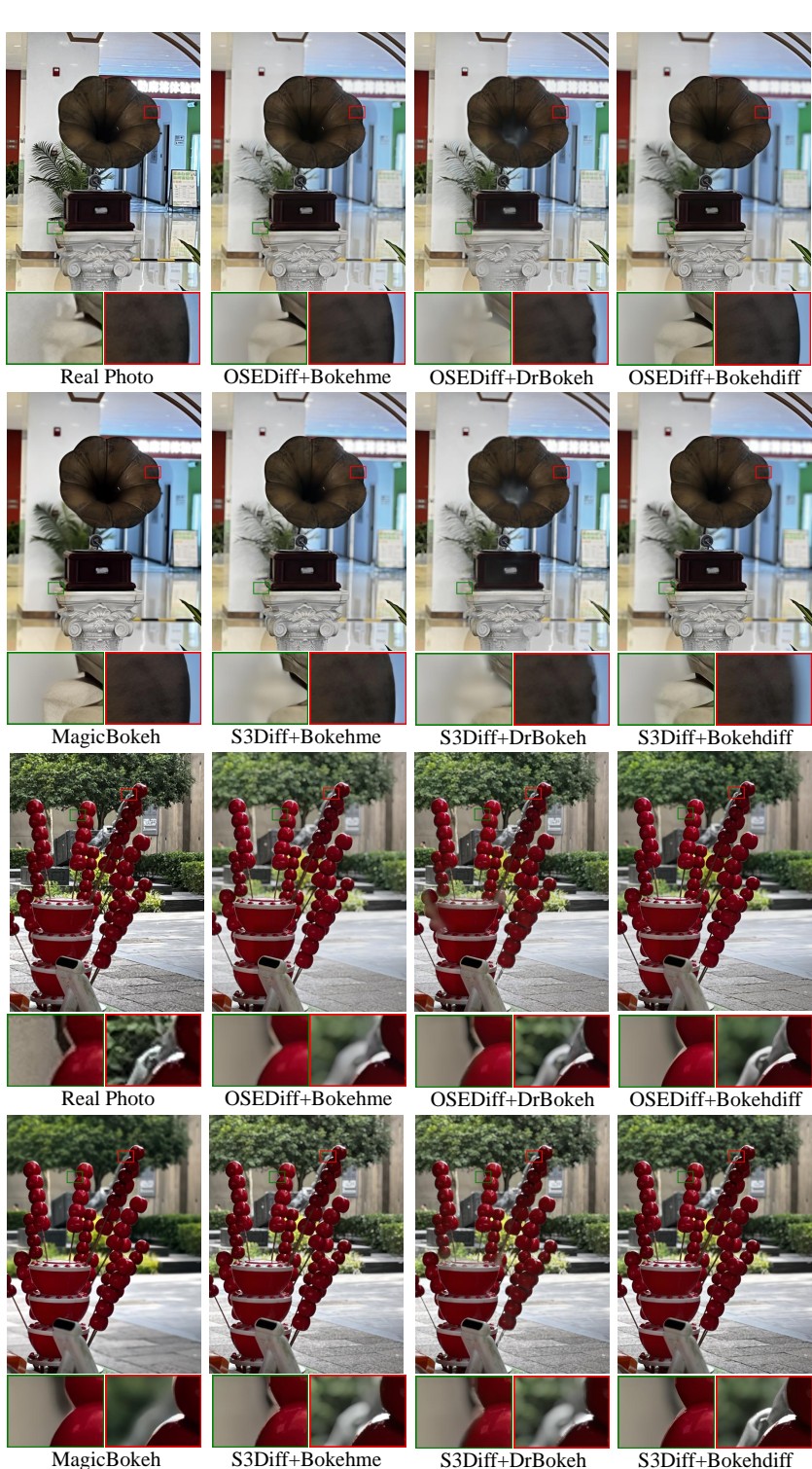

Figure 14: Qualitative results on 5x zoom images captured with an iPhone 13.

