# OpenReview forum: "Diffusion-Based Photorealistic Bokeh Rendering for Mobile Devices"
_ICLR.cc/2026/Conference — ICLR 2026 Conference Withdrawn Submission_

### Official Review · Reviewer_zWMR · 2025-10-24

**Soundness:** 1
**Presentation:** 2
**Contribution:** 2
**Rating:** 2
**Confidence:** 4

**Summary:**

This paper introduces "MagicBokeh," a unified, diffusion-based framework for rendering photorealistic bokeh effects on images captured with high digital zoom on mobile devices. The authors argue that current bokeh methods perform poorly on such low-quality, all-in-focus inputs, and that a naive two-stage approach (super-resolution followed by bokeh rendering) suffers from issues like error accumulation. MagicBokeh aims to solve this by jointly optimizing both tasks in a single model, using an alternative training strategy and a focus-aware attention mechanism.

**Strengths:**

This paper design a unified framework for a computational photography application with the help of Diffusion Models.

**Weaknesses:**

1. On the Significance and Timeliness of the Problem Formulation:
My primary concern is whether the problem this paper aims to solve is significant and relevant in the current technological landscape. The central argument rests on the premise that bokeh rendering for low-quality, high-digital-zoom images is a critical, unsolved issue. I am not fully convinced of this for the following reasons:
Relevance of the Motivating Scenario: The paper uses an iPhone 13 at 5x digital zoom as a key example to demonstrate the low-quality input problem (Supplementary, Fig. 14). This hardware was released four years prior to the target publication year (2026). Contemporary flagship smartphones increasingly integrate multiple, high-quality optical telephoto lenses, which significantly reduce the reliance on aggressive digital zoom. Motivating a novel method based on a scenario that is arguably becoming less common due to hardware advancements makes the problem statement feel somewhat dated. The paper needs to better establish that this specific setting remains a persistent and widespread challenge, rather than an artifact of older hardware.
Evidence of the "Error Accumulation" Problem: The paper claims that a two-stage Super-Resolution (SR) + Bokeh pipeline leads to significant error accumulation. However, the evidence for this claim is not compelling.
Quantitative Results: The results in Table 1 do not show a significant advantage for MagicBokeh. The performance differences in PSNR, SSIM, and other metrics between the proposed unified model and the two-stage baselines (e.g., OSEDiff + BokehDiff) are marginal. If error accumulation were a critical flaw in the two-stage approach, one would expect to see a much more substantial performance gap.
Qualitative Results: The qualitative comparisons in Figure 14 are similarly unconvincing. At a fine-grained level, it is difficult to perceive a clear and consistent superiority of MagicBokeh over the SR + Bokeh baselines. The improvements are subtle at best.
In essence, the work appears to be solving a problem that is neither clearly defined as critical nor strongly demonstrated through the paper's own experiments.

2. On the Justification and Impact of the Contribution:
Stemming from the above, the overall contribution feels incremental rather than fundamental. The work is framed as a solution to a clear-cut problem (error accumulation), but the solution itself does not deliver a clear-cut improvement. This suggests that either the problem is not as severe as claimed, or the proposed solution is not as effective as needed.
For a top-tier conference, research is expected to either tackle a significant, well-established problem with a novel solution, or to convincingly identify and solve a previously under-appreciated but important problem. This work, in its current state, struggles on both fronts. It appears to be an engineering exercise on a problem of questionable significance, with results that are not strong enough to justify its claims.

**Questions:**

See weaknesses.

---

### Official Review · Reviewer_M4FB · 2025-10-26

**Soundness:** 2
**Presentation:** 3
**Contribution:** 2
**Rating:** 4
**Confidence:** 4

**Summary:**

In this paper, the authors propose MagicBokeh, a unified diffusion-based framework that improves image quality and bokeh effects. Specifically, they present a alternative training strategy and focus-aware mask attention module. They further optimize depth estimation on low-quality images by degradation-aware depth module. My main concern is that the proposed method is not validated on any mobile devices.

**Strengths:**

- Alternating training alleviates conflicts between ISR and bokeh, the ablation shows an improvement in the quality of background blur.
- Clearly pointing out the challenges of image degradation (noise, blurred boundaries) caused by high magnification digital zoom to existing bokeh methods has practical application value.

**Weaknesses:**

- Mobile claims and efficiency evaluation. This paper emphasizes mobile devices, but has not been validated on mobile devices. In addition, only GPU inference time is reported (Table 1), but parameters/FLOPs/memory are not reported. The memory usage of Depth anything V2 and diffusion model should be very large, and the authors should discuss this.
- Inference time. In table 1,the inference time of the proposed method is faster than other non diffusion models, such as Bokehme. But the paper did not mention the optimization and improvement of their method in terms of time consuming, how was this achieved.
- Baseline comparison is unfair: the two-stage method uses Depth Anything v2 to estimate depth, while MagicBokeh uses self-developed DA depth. If the DA depth itself is better than the baseline, the performance gain may be attributed to depth estimation rather than a unified architecture.

**Questions:**

- The robustness of this method. Although the paper proposes the DA depth module, the robustness of the subsequent network should be verified. The authors should compare their method with degraded depth maps and without degraded depth maps.
- Weak improvement for DA depth. Tab. 2 shows that there is almost no difference in PSNR/LPIPS between the "w/o DA depth" and the full model.
- Can the focus position and blur size be controlled? If set to no blur, does the super-resolution effect of this method work better than other methods.
- How does this method perform when the edge of the object is human and animal hair？

---

### Official Review · Reviewer_2Wwe · 2025-10-27

**Soundness:** 3
**Presentation:** 2
**Contribution:** 3
**Rating:** 4
**Confidence:** 3

**Summary:**

This paper proposes MagicBokeh, a unified diffusion-based framework for photorealistic bokeh rendering on mobile devices, specifically addressing the challenges of high digital zoom images with degraded quality. Unlike traditional two-stage pipelines that separately perform super-resolution and bokeh rendering, MagicBokeh jointly optimizes both tasks in a single-step architecture through an alternative training strategy, focus-aware mask attention, and a degradation-aware depth module. Experiments on both synthetic and real-world datasets demonstrate that MagicBokeh achieves state-of-the-art performance.

**Strengths:**

1. This paper proposes a unified diffusion-based framework for simultaneous Real-ISR and photorealistic bokeh rendering on mobile devices.
2. Compared with traditional two-stage pipelines, MagicBokeh achieves superior visual quality and significantly faster inference speed, demonstrating its effectiveness and efficiency for real-world mobile bokeh rendering.

**Weaknesses:**

1. In FAMA and Eq. 2, the role of the binary mask in attention reweighting is unclear. When I first read this section, I thought the mask output would be integrated into the loss computation, but it turned out not to be — which made me quite confused. And there is no visualization or attention map analysis to support that FAMA indeed enhances focus fidelity.
2. What would be the performance of the methods listed in Table 1 if they were retrained on the LR+bokeh dataset? Could their results possibly surpass those of MagicBokeh under such training conditions?
3. In Table 2, the performance gains from individual components (FAMA, Strategy, DA Depth) are marginal. These small gains might fall within the variance of random initialization. The authors should perform multiple runs with different random seeds and report standard deviations to verify statistical significance.
4. Although the paper claims that the framework is designed for mobile devices, it lacks on-device performance evaluation, as all efficiency analyses are conducted solely on a L40s GPU.
5. Are the results robust for photos captured by Samsung Galaxy Phone, Google Pixel, and iPhone 17 devices?
6. The paper presents a well-motivated research problem and an overall solid architectural design; however, based on the experimental results, the proposed modules appear to have little to no actual impact on performance.

**Questions:**

See weaknesses.

---

### Official Review · Reviewer_qgXT · 2025-10-28

**Soundness:** 2
**Presentation:** 2
**Contribution:** 2
**Rating:** 4
**Confidence:** 4

**Summary:**

This paper presents MagicBokeh, a unified diffusion-based framework for photorealistic bokeh rendering on mobile devices, particularly addressing challenges under high digital zoom conditions. Unlike traditional two-stage pipelines (super-resolution → bokeh rendering), MagicBokeh performs joint Real-ISR and bokeh rendering in a single-step architecture.

**Strengths:**

1.	Integrates super-resolution and bokeh rendering into a single diffusion model, improving efficiency and avoiding error accumulation between stages.
2.	Better performance on user study from human perspective.

**Weaknesses:**

1.Questionable suitability for mobile devices.

The paper’s title emphasizes "for mobile devices", yet no computational complexity, FLOPs, or parameter count is provided. Diffusion-based models are typically heavy, and even though inference is faster than two-stage pipelines, the real computational cost is likely much higher than classical or CNN-based methods such as BokehMe or Dr.Bokeh. There is no discussion or measurement of feasibility on actual mobile hardware.

2.Lack of efficiency analysis.

The paper repeatedly claims that the proposed method has computational efficiency, yet no quantitative evidence is provided to support this statement. Since efficiency is presented as one of the main advantages of the method, the authors should include a quantitative comparison of parameter counts and computational costs between the proposed approach and other competing methods to substantiate this claim.

3.Limited diversity and scale of evaluation datasets.

The EBB400 benchmark only contains cases with foreground-focused, fixed blur intensity, limiting diversity in focus settings and bokeh intensities. Furthermore, only 400 samples are used, which is too small for a diffusion-based model evaluation. The authors mention building a synthetic dataset via ray tracing, yet no quantitative evaluation is reported on that dataset.

4.Potential over-synthesis and lack of fidelity analysis on HQ inputs.

It remains unclear how the model behaves when the input is already high-quality. Diffusion models may introduce extra hallucinated details or artifacts, degrading fidelity. The paper does not discuss or test this aspect—for example, evaluating on the original non-degraded EBB dataset (without simulated degradations) to assess consistency with ground-truth images.

5.Insuffienct novelty.

The paper lacks sufficient novelty. The idea of HQ Feature Extraction is derived from existing works; the Controllable Bokeh Rendering Module largely follows the framework of ControlNet with minimal modification; and the mask attention mechanism is a well-established technique without notable innovation.

6. Writing issues.

The paper contains numerous issues in writing, unclear expressions, and improper formatting, which make it difficult to read. For example: in line 102, Depth Anything V2 is mentioned for the first time but lacks a citation; in line 131, DeepFocus Senaras et al. (2018) should be written as DeepFocus (Senaras et al., 2018); in line 133, Dr.Bokeh Sheng et al. (2024) should be Dr.Bokeh (Sheng et al., 2024); in line 178, controlnet should be capitalized as ControlNet; in line 213, the SR Bokeh dataset is mentioned without prior explanation, and the meaning only becomes clear in Section 4; in line 234, bokeh conditions appears for the first time without context, making it difficult to understand what it refers to.

**Questions:**

1.	What is the actual model size (parameters, FLOPs) and runtime on typical mobile? Could the authors release a lightweight version or distillation variant for actual mobile deployment?

2.	How does the model perform when given already high-resolution, clean inputs? Does it over-smooth or hallucinate new details?

3.	Why was the synthetic ray-traced bokeh dataset not used for quantitative testing, despite being carefully constructed? Can the authors provide results under different focal-plane settings or variable blur intensity, beyond the fixed-focus EBB400 benchmark?

---

### Note · Authors · 2025-11-12

I have read and agree with the venue's withdrawal policy on behalf of myself and my co-authors.